# Leveraging Behavioral and Social Information for Weakly Supervised Collective Classification of Political Discourse on Twitter

## Abstract

Framing is a political strategy in which politicians carefully word their statements in order to control public perception of issues. Previous works exploring political framing typically analyze frame usage in longer texts, such as Congressional speeches. We present a collection of weakly supervised models which harness collective classification to predict the frames used in political discourse on the microblogging platform, Twitter. Our global probabilistic models show that by combining both lexical features of tweets and network-based behavioral features of Twitter, we are able to increase the average, unsupervised $F_1$ score by 21.52 points over a lexical baseline alone.

## 1 Introduction

The importance of understanding political discourse on social media platforms is becoming increasingly clear. In recent U.S. presidential elections, Twitter was widely used by all candidates to promote their agenda, interact with supporters, and attack their opponents. Social interactions on such platforms allow politicians to quickly react to current events and gauge interest in and support for their actions. These dynamic settings both emphasize the importance of constructing automated tools for analyzing this content, but also the difficulty of constructing such tools as the language used to discuss new events and political agendas continuously changes. Consequently, the rich social interactions on Twitter can be leveraged to help support such analysis by providing alternatives to direct supervision.

In this paper we focus on political framing, a very nuanced political discourse analysis task, on Twitter, a relatively unexplored domain for this task. Framing (Entman, 1993; Chong and Druckman, 2007) is employed by politicians to bias the discussion towards their stance by emphasizing specific aspects of the issue. For example, the debate around increasing the minimum wage can be framed as a *quality of life* issue or as an *economic* issue. While the first frame supports increasing minimum wage because it betters workers' lives, the second frame, by conversely emphasizing the costs involved, opposes the increase. Using framing to analyze political discourse has gathered significant interest over the last few years (Tsur et al., 2015; Card et al., 2015; Baumer et al., 2015) as a way to automatically analyze political discourse in Congressional speeches and political news articles. Our dataset consists of the tweets authored by all members of the U.S. Congress from both parties, dealing with several policy issues (e.g., immigration, ACA, etc.). We annotated these tweets by adapting the annotation guidelines developed by Boydstun et al. for Twitter. More details about the annotation process are provided in Section 3.

Twitter issue framing is a challenging multi-label prediction task. Each tweet can be labeled as using one or more frames, out of 17 possibilities, while only providing 140 characters as input to the classifier. Instead of following a supervised path, our main goal in this paper is to evaluate whether the *social and behavioral information* available on Twitter is sufficient for constructing a reliable classifier for this task. We approach this task using a weakly supervised collective classification approach which leverages the dependencies between tweet frame predictions based on the interactions between their authors.

We model these dependencies by connecting Twitter users that have social connections or behavioral similarity. Social connections are di-

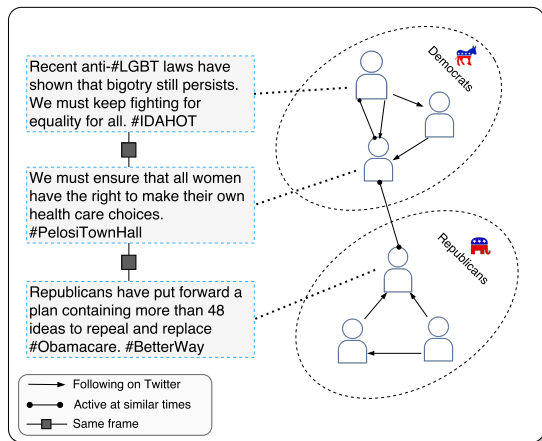

Figure 1: Collective classification framework models the dependency between tweet frame predictions and social interactions and behavioral similarities between their authors.

rected dependencies that represent the followers of each user and retweeting behavior (i.e., user A retweets user B's content). Interestingly, these social connections capture the flow of influence within political parties; however, the number of connections that cross party lines is extremely low. Instead we rely on capturing behavioral similarity between users to provide us with this information. We construct a temporal histogram for each politician which captures their Twitter activity over time. Users whose Twitter activity peaks at similar times tend to discuss issues in similar ways, making the comparison between their frames usage easier. Figure 1 shows an example of such a network and the prediction dependencies it forms. In addition to the edges, we also represent each politician's party affiliation and the frequent phrases (e.g., bigrams or trigrams) used by politicians on Twitter.

We compile these structural dependencies into a graphical model using Probabilistic Soft Logic (PSL), a recently introduced probabilistic modeling framework [1]. As described in Section 4, PSL combines these aspects declaratively by specifying high level rules over a relational representation of tweet features. The rules are compiled into a graphical model called a hinge-loss Markov random field (Bach et al., 2013), which is used to make the frame prediction. Instead of direct supervision we take a bootstrapping approach by providing a small seed set of keywords adapted from Boydstun et al., for each frame.

Our experiments show that modeling the social and behavioral connections improves $F_1$ prediction scores in both supervised and unsupervised settings, with double the increase in the latter. We apply our unsupervised model to our entire tweets dataset to analyze framing patterns over time by both party and individual politicians. Our analysis provides insight into the usage of framing for identification of *aisle-crossing politicians*, i.e., those politicians who vote against their party.

## 2 Related Work

Issue framing is related to the broader challenges of biased language analysis (Recasens et al., 2013; Choi et al., 2012; Greene and Resnik, 2009) and subjectivity (Wiebe et al., 2004). Several previous works have explored framing in public statements, Congressional speeches, and news articles (Fulgoni et al., 2016; Tsur et al., 2015; Card et al., 2015; Baumer et al., 2015). Our approach builds upon the previous work on frame analysis of Boydstun et al., by adapting and applying their annotation guidelines for Twitter.

In recent years there has been growing interest in analyzing political discourse. Most previous work focuses on opinion mining and stance prediction (Sridhar et al., 2015; Hasan and Ng, 2014; Abu-Jbara et al., 2013; Walker et al., 2012; Abbott et al., 2011; Somasundaran and Wiebe, 2010, 2009). Analyzing political tweets has also attracted considerable interest: a recent SemEval task looked into stance prediction[2], and more related to our work, Tan et al. have shown how wording choices can affect message propagation on Twitter. Two recent works look into predicting stance (at user and tweet levels respectively) on Twitter using PSL (Johnson and Goldwasser, 2016; Ebrahimi et al., 2016). Other works focus on identifying and measuring political ideologies (Iyyer et al., 2014; Bamman and Smith, 2015; Sim et al., 2013), policies (Nguyen et al., 2015), and voting patterns (Gerrish and Blei, 2012).

Exploiting social interactions and group structure for prediction has also been explored (Sridhar et al., 2015; Abu-Jbara et al., 2013; West et al., 2014). Works focusing on inferring signed social networks (West et al., 2014), stance classification (Sridhar et al., 2015), social group modeling (Huang et al., 2012), and collective classification using PSL (Bach et al., 2015) are closest

---

[1] http://psl.cs.umd.edu

[2] http://alt.qcri.org/semeval2016/task6/

to our approach. Unsupervised and weakly supervised models of Twitter data for several various tasks have been suggested, including: profile (Li et al., 2014b) and life event extraction (Li et al., 2014a), conversation modeling (Ritter et al., 2010), and methods for dealing with the unique language used in microblogging platforms (Eisenstein, 2013).

Predicting political affiliation and other characteristics of Twitter users has been explored (Volkova et al., 2015, 2014; Yano et al.; Conover et al., 2011). Other works have focused on sentiment analysis (Pla and Hurtado, 2014; Bakliwal et al., 2013), predicting ideology (Djemili et al., 2014), automatic polls based on Twitter sentiment and political forecasting using Twitter (Bermingham and Smeaton, 2011; O'Connor et al., 2010; Tumasjan et al., 2010), as well as distant supervision applications (Marchetti-Bowick and Chambers, 2012).

## 3   Data Collection and Annotation

We collected 184,914 of the most recent tweets of members of the U.S. Congress (both the House of Representatives and Senate). Using an average of ten keywords per issue, we filtered out tweets not related to the following six issues of interest: (1) abortion access, (2) the Affordable Care Act (i.e., ACA or Obamacare), (3) gun rights versus gun control, (4) immigration policies, (5) acts of terrorism, and (6) issues concerning the LGBTQ community. Forty politicians (10 Republicans and Democrats, from both the House and Senate), were chosen randomly for annotation.

Two graduate students were trained to annotate each tweet with a frame using the Policy Frames Codebook developed by Boydstun et al.. Brief descriptions of each frame are given in Table 1. During the annotation process, we found that Boydstun's Frame 15 (Other) was never used. We therefore dropped it from this study. In its place, we propose 3 Twitter-specific frames: Factual (15), Promotion (16), and Personal Sympathy and Support (17). Due to the compound nature of tweets and the consequent possibility of multiple frames per tweet (also discussed in Card et al.), annotators were allowed to label each tweet with multiple frames when one primary frame was not possible. For all such tweets, annotators repeated the annotation process together to determine if the tweets could be represented by a single frame or required

more. We computed the inter-annotator agreement using Cohen's Kappa statistic and have an agreement of 73.4%, which can be viewed as proof of the difficulty of frame classification for tweets. Table 2 presents the statistics of our tweets dataset, which will be released for the community's use.

## 4   Global Models of Twitter Language and Activity

Due to the dynamic nature of political discourse on Twitter, we design weakly supervised PSL models to require as little supervision as possible. The only sources of supervision our approach requires include: unigrams related to the issues, unigrams adapted from the Boydstun et al. Codebook for frames, and political party of the author of the tweets[3]. The local models described in this section are data-dependent and used to extract and format information from tweets into input for PSL predicates and rules. Our goal to label each tweet with a frame can be defined in PSL notation as a *target predicate*: FRAME(T, F), where T represents a tweet, and F represents one of the 17 frames listed in Table 1.

### 4.1   Global Modeling Using PSL

PSL is a declarative modeling language which can be used to specify weighted, first-order logic rules. These rules are compiled into a hinge-loss Markov random field which defines a probability distribution over possible continuous value assignments to the random variables of the model (Bach et al., 2015) [4]. This probability density function is represented as:

$$P(\mathbf{Y} \mid \mathbf{X}) = \frac{1}{Z} \exp\left( - \sum_{r=1}^{M} \lambda_r \phi_r(\mathbf{Y}, \mathbf{X}) \right)$$

where $Z$ is a normalization constant, $\lambda$ is the weight vector, and

$$\phi_r(\mathbf{Y}, \mathbf{X}) = (\max\{l_r(\mathbf{Y}, \mathbf{X}), 0\})^{\rho_r}$$

is the hinge-loss potential specified by a linear function $l_r$. The exponent $\rho_r \in 1, 2$ is optional. Each potential represents the instantiation of a rule, which takes the following form:

$$\lambda_1 : P_1(x) \wedge P_2(x, y) \to P_3(y)$$
$$\lambda_2 : P_1(x) \wedge P_4(x, y) \to \neg P_3(y)$$

$P_1, P_2, P_3$, and $P_4$ are predicates (e.g., political party, issue, frame, and presence of n-grams) and $x, y$ are variables. Each rule has a weight $\lambda$ which reflects that rule's importance and is learned using

---

[3]All information will be released with our dataset at: www.\*\*\*.\*\*\*.

[4]This is the opposite of other probabilistic logical models, e.g. MLNs, in which rules are strictly true or false.

| FRAME NUMBER, FRAME NAME, AND BRIEF DESCRIPTION OF FRAME |
|---|
| 1. ECONOMIC: *Pertains to the economic impacts of a policy* |
| 2. CAPACITY & RESOURCES: *Pertains to lack of or availability of resources* |
| 3. MORALITY & ETHICS: *Motivated by religious doctrine, righteousness, sense of responsibility* |
| 4. FAIRNESS & EQUALITY: *Of how laws, punishments, resources, etc. are distributed among groups* |
| 5. LEGALITY, CONSTITUTIONALITY, & JURISDICTION: *Including court cases, restriction and expressions of rights* |
| 6. CRIME & PUNISHMENT: *Policy violation and consequences* |
| 7. SECURITY & DEFENSE: *Threats or defenses/preemptive actions to protect against threats* |
| 8. HEALTH & SAFETY: *Includes care access and effectiveness* |
| 9. QUALITY OF LIFE: *Effects on individual and community life* |
| 10. CULTURAL IDENTITY: *Culture's norms, trends, customs* |
| 11. PUBLIC SENTIMENT: *Pertains to opinions, polling, and demographics* |
| 12. POLITICAL FACTORS & IMPLICATIONS: *Efforts, stances, filibusters, lobbying, maneuvering, references to other politicians* |
| 13. POLICY DESCRIPTION, PRESCRIPTION, & EVALUATION: *Discussion about effectiveness of current or proposed policies* |
| 14. EXTERNAL REGULATION AND REPUTATION: *Interstate and international relationships of the U.S.* |
| 15. FACTUAL: *Expresses a pure fact, with no political spin* |
| 16. PROMOTION: *Promotes another person or the author in some way, e.g. television appearances* |
| 17. PERSONAL SYMPATHY & SUPPORT: *Expresses sympathy, emotional response, or solidarity with others* |

Table 1: Frames of Boydstun. The first 14 are Boydstun's frames and the last 3 are our proposed Twitter-specific frames. Boydstun's original Frame 15 (Other) is omitted from this study.

| Tweets | BY PARTY | | BY ISSUE | | | | | |
|---|---|---|---|---|---|---|---|---|
| | REPUBLICAN | DEMOCRAT | ABORTION | ACA | GUNS | IMMIGRATION | TERRORISM | LGBTQ |
| ENTIRE DATASET | 48504 | 43953 | 6467 | 35854 | 15532 | 13442 | 15205 | 6046 |
| LABELED SUBSET | 894 | 1156 | 170 | 564 | 543 | 233 | 446 | 183 |

Table 2: Statistics of Collected Tweets.

the Expectation-Maximization algorithm in our unsupervised experiments. Using concrete constants *a, b* (e.g., tweets and words) which instantiate the variables $x, y$, model atoms are mapped to continuous [0,1] assignments. More important rules (i.e., those with larger weights) are given preference by the model.

## 4.2 Language Based Models

**Unigrams:** Using the guidelines provided in the Policy Frames Codebook (Boydstun et al., 2014), we adapted a list of expected unigrams for each frame. For example, unigrams that should be related to Frame 12 (Political Factors & Implications) include: filibuster, lobby, Democrats, Republicans. We expect that if a tweet and frame contain a matching unigram, then that tweet is likely expressed by that frame. The information that tweet T has expected unigram U of frame F is represented with the PSL predicate: HASUNIGRAM$_F$(T, U). This knowledge is then used as input to PSL Model 1 via the rule: HASUNIGRAM$_F$(T, U) →FRAME(T, F) (shown in line 1 of Table 3).

However, not every tweet will have a unigram that matches those in this list. Under the intuition that at least one unigram in a tweet should be *similar* to a unigram in the list, we designed the fol-

lowing *MaxSim* metric to compute the maximum similarity between a word in a tweet and a word from the list of unigrams.

$$\text{MAXSIM(T, F)} = \underset{u \in F, w \in T}{\arg\max} \text{SIMILARITY(W,U)}$$

(1)

T is a tweet, W is each word in T, and U is each unigram in the list of expected unigrams (per frame). Similarity is the computed `word2vec` similarity of each word in the tweet with every unigram in the list of unigrams for each frame. The frame F of the maximum scoring unigram is input to the PSL predicate: MAXSIM$_F$(T, F), which indicates that tweet T has the highest similarity to frame F.

**Bigrams and Trigrams:** In addition to unigrams, we also explored the effects of political party *slogans* on frame prediction. Slogans are common catch phrases or sayings that people typically associate with different U.S. political parties. For example, Republicans are known for using "repeal and replace", a trigram, when they discuss the ACA. Similarly, in the 2016 U.S. presidential election, Secretary Hillary Clinton's campaign slogan became "Love Trumps Hate". To visualize slogan usage by parties for different issues, we used the *entire* tweets dataset, including all unlabeled tweets, to extract the top bigrams and trigrams per party for each issue. The

| Types of Models | Model Number | Basis of Model | Example of PSL Rules |
|---|---|---|---|
| Language Based | 1 | Unigrams | $\text{HASUNIGRAM}_F$(T, U) →Frame(T, F) |
| | 2 | Bigrams | $\text{HASUNIGRAM}_F$(T, U) ∧$\text{PARTYBIGRAM}_P$(T, B) →Frame(T, F) |
| | 3 | Trigrams | $\text{HASUNIGRAM}_F$(T, U) ∧$\text{PARTYTRIGRAM}_P$(T, TG) →Frame(T, F) |
| Behavior Based | 4 | Temporal Activity | SameTime(T1, T2) ∧Frame(T1, F) →Frame(T2, F) |
| | 5 | Retweet Patterns | Retweets(T1, T2) ∧Frame(T1, F) →Frame(T2, F) |
| | 6 | Following Network | Follows(T1, T2) ∧Frame(T1, F) →Frame(T2, F) |

Table 3: Examples of PSL Model Rules. Each model adds to the rules of the previous model. For brevity we show a subset of the rules and omit full model combinations. The full list of rules per model will be released with our dataset.

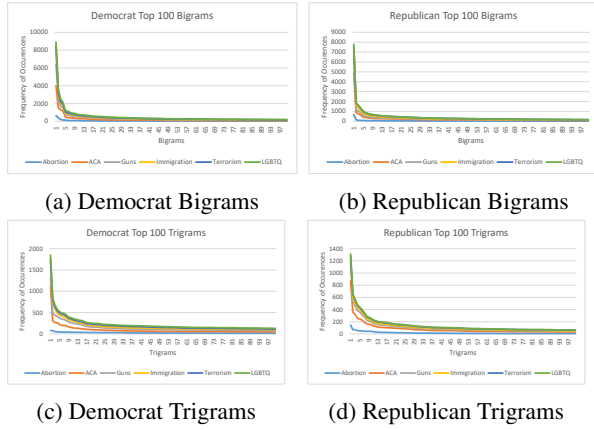

(a) Democrat Bigrams (b) Republican Bigrams

(c) Democrat Trigrams (d) Republican Trigrams

Figure 2: Distributions of Bigrams and Trigrams by Party.

histograms in Figure 2 show these distributions for the top 100 bigrams and trigrams. Based on these results, we use the top 20 bigrams (e.g., *women's healthcare* and *immigration reform*) and trigrams (e.g. *prevent gun violence*) as input to PSL predicates $\text{PARTYBIGRAM}_P$(T, B) and $\text{PARTYTRIGRAM}_P$(T, TG). These rules represent that tweet T has bigram B or trigram TG from the respective phrase lists of either party (i.e., P represents either Democrat or Republican in the rule instantiation).

### 4.3 Twitter Behavior Based Models

In addition to language based features of tweets, we also exploit the behavioral features of Twitter including similarities between temporal activity and network relationships.

**Temporal Similarity:** When an event happens politicians are most likely to tweet about that event within hours of its occurrence. Similarly, most politicians tweet about the event most frequently the day of the event and this frequency decreases over time. We expect that the frames used the day of an event will be similar and change over time. To capture this behavior we use the PSL predicate SameTime(T1, T2). This indicates that

tweet T1 occurs around the same time as tweet T2[5]. This information is used in Model 4 via rules such as: SameTime(T1, T2) & Frame(T1, F) →Frame(T2, F), as shown in line 4 of Table 3.

**Network Similarity:** Finally, we expect that politicians who share ideologies, and thus are likely to frame issues similarly, will retweet and/or follow each other on Twitter. Due to the compound nature of tweets, retweeting with additional comments can add more frames to the original tweet. Additionally, politicians on Twitter are more likely to follow members of their own party or similar non-political entities than those of the opposing party. To capture this network-based behavior we use two PSL predicates: Retweets(T1, T2) and Follows(T1, T2). These predicates indicate that the content of tweet T1 includes a retweet of tweet T2 and that the author of T1 follows the author of T2 on Twitter, respectively. The last two lines of Table 3 show examples of how network similarity is incorporated into PSL rules.

## 5 Experiments

**Experimental Settings:** We provide an analysis of our PSL models under both supervised and unsupervised settings. As shown in Table 4, Model 1 in a supervised setting is similar to the traditional bag-of-words baseline, which we use as our baseline to improve upon. Additionally, Model 6 of the supervised, collective network setting represents the best results we can achieve. In the supervised experiments, we used five-fold cross validation with randomly chosen splits.

We also explore the results of our PSL models in an unsupervised setting because the highly dynamic nature of political discourse on Twitter makes it unrealistic to expect annotated data to

---

[5]We conducted experiments with different hour and day limits and found that using a time frame of one hour results in the best accuracy.

| SETTING | UNIGRAM BASELINE | COLLECTIVE NETWORK |
|---|---|---|
| SUPERVISED | 66.02 | 77.79 |
| UNSUPERVISED | 37.14 | 58.66 |

Table 4: Baseline and Skyline Weighted Averages. Supervised, unigram model is the lowest score the models can achieve. Supervised, collective network model is the highest.

generalize to future discussions. The only source of supervision comes from the initial unigrams lists as described in Section 4. The labeled tweets are used for evaluation only. As seen in Table 4, we are able to improve the unsupervised model to within an $F_1$ score of 7.36 points of the unigram baseline, and 19.13 points of the best supervised score.

**Evaluation Metrics:** Since each tweet can have more than one frame, our prediction task is a multilabel classification task. The precision of a multilabel model is the ratio of how many predicted labels are correct:

$$Precision = \frac{1}{T} \sum_{t=1}^{T} \frac{|Y_t \cap h(x_t)|}{|h(x_t)|} \qquad (2)$$

The recall of this model is the ratio of how many of the actual labels were predicted:

$$Recall = \frac{1}{T} \sum_{t=1}^{T} \frac{|Y_t \cap h(x_t)|}{|Y_t|} \qquad (3)$$

In both formulas, T is the number of tweets, $Y_t$ is the true label for tweet $t$, $x_t$ is a tweet example, and $h(x_t)$ are the predicted labels for that tweet. The $F_1$ score is computed as the harmonic mean of the precision and recall.

**Analysis of Supervised Experiments:** Table 5 shows the results of our supervised experiments. Here we can see that by adding Twitter behavior (beginning with Model 4), our behavior-based models achieve the best $F_1$ scores across all frames. Model 4 achieves the highest results on two frames, suggesting retweeting and network follower information does not help improve the prediction score for these frames. Model 5 similarly achieves the highest prediction for 5 of the frames, suggesting network follower information cannot further improve the score for these frames. Overall, the Twitter behavior based models are able to outperform language based models alone, including the best performing language

model (Model 3) which combines unigrams, bigrams, and trigrams together to collectively infer the correct frame(s).

**Analysis of Unsupervised Experiments:** In the unsupervised setting, Model 6, the combination of language and Twitter behavior features achieves the best results on 16 of the 17 issues, as shown in Table 6. There are a few interesting aspects of the unsupervised setting which differ from the supervised setting. Six of the frame predictions do worse in Model 2, which is double that of the supervised version. This is likely due to the presence of overlapping bigrams across frames and issues, e.g., "women's healthcare" could appear in both Frames 4 and 8 and the issues of ACA and abortion. However, all six are able to improve with the addition of trigrams (Model 3), whereas only 1 of 3 frames improves in the supervised setting. This suggests that bigrams may not be as useful as trigrams in an unsupervised setting. Finally, in Model 5, which adds retweet behaviors, we notice that 5 of the frames decrease in $F_1$ score and 11 of the frames have the same score as the previous model. These results suggest that retweet behaviors are not as useful as the follower network relationships in an unsupervised setting.

## 6 Qualitative Analysis

To explore the usefulness of frame identification in political discourse analysis, we apply our best performing model (Model 6) on the *unlabeled* dataset to determine framing patterns over time, both by party and individual. Figure 3 shows the results of our frame analysis over time for two issues: ACA and terrorism[6]. We compiled the predicted frames for tweets from 2014 to 2016 for each party. Figure 4 presents the results of frame prediction for 2015 tweets of individuals for these two issues.

**Party Frames:** From Figure 3(a) we can see that Democrats mainly use Frames 1, 4, 8, 9, and 15 to discuss ACA, while Figure 3(c) shows that Republicans predominantly use Frames 1, 8, 9, 12, and 13. Though the parties use similar frames, they are used to express different agendas. For example, Democrats use Frame 8 to indicate the positive effect that the ACA has had in granting more Americans health care access. Republicans,

---

[6]Due to space, we omit the other 4 issues. These 2 were chosen because they are among the most frequently discussed topics in our dataset.

| Frame Number | Frame | RESULTS OF SUPERVISED PSL MODEL FRAME PREDICTIONS | | | | | |
|---|---|---|---|---|---|---|---|
| | | MODEL 1 | MODEL 2 | MODEL 3 | MODEL 4 | MODEL 5 | MODEL 6 |
| 1 | ECONOMIC | 85.19 | 85.19 | 86.73 | 87.72 | 87.72 | **89.88** |
| 2 | CAPACITY & RESOURCES | 55.38 | 61.54 | 76.71 | 77.11 | 77.11 | **79.55** |
| 3 | MORALITY | 73.39 | 80.52 | 86.95 | 87.5 | **87.43** | **87.43** |
| 4 | FAIRNESS | 63.56 | 67.83 | 65.19 | 69.91 | 79.53 | **82.35** |
| 5 | LEGALITY | 80.41 | 80.78 | 80.79 | **83.33** | 81.79 | 82.16 |
| 6 | CRIME | 54.55 | 54.55 | 66.67 | **76.92** | **76.92** | **76.92** |
| 7 | SECURITY | 84.40 | 82.14 | 84.10 | 86.67 | 86.67 | **88.48** |
| 8 | HEALTH | 73.50 | 75.76 | 75.59 | 77.46 | **79.71** | **79.71** |
| 9 | QUALITY OF LIFE | 69.39 | 68.00 | 69.39 | 72.34 | 72.34 | **82.93** |
| 10 | CULTURAL | 75.86 | 78.57 | 81.25 | 81.25 | 81.25 | **85.71** |
| 11 | PUBLIC SENTIMENT | 12.25 | 15.25 | 24.62 | 24.24 | 26.24 | **29.41** |
| 12 | POLITICAL | 54.21 | 63.31 | 74.33 | 74.42 | **74.52** | **74.52** |
| 13 | POLICY | 55.75 | 58.87 | 60.25 | 61.54 | 64.06 | **65.06** |
| 14 | EXTERNAL REGULATION | 60.71 | 59.15 | 64.71 | 74.35 | 74.35 | **85.71** |
| 15 | FACTUAL | 66.56 | 68.00 | 71.43 | 81.82 | 80.82 | **82.85** |
| 16 | PROMOTION | 85.71 | 86.46 | 86.58 | 87.34 | 87.33 | **91.76** |
| 17 | PERSONAL | 71.79 | 71.71 | 74.73 | 75.00 | **77.55** | **77.55** |
| | WEIGHTED AVERAGE | 66.02 | 68.78 | 72.49 | 74.40 | 75.71 | **77.79** |

Table 5: $F_1$ Scores of Supervised PSL Models. The highest prediction per frame is marked in bold.

| Frame Number | Frame | RESULTS OF UNSUPERVISED PSL MODEL FRAME PREDICTIONS | | | | | |
|---|---|---|---|---|---|---|---|
| | | MODEL 1 | MODEL 2 | MODEL 3 | MODEL 4 | MODEL 5 | MODEL 6 |
| 1 | ECONOMIC | 31.82 | 31.52 | 69.57 | 72.22 | 72.22 | **73.23** |
| 2 | CAPACITY & RESOURCES | 23.38 | 28.51 | 40.00 | **41.18** | **41.18** | **41.18** |
| 3 | MORALITY | 28.63 | 29.41 | 47.67 | 53.98 | 43.06 | **53.99** |
| 4 | FAIRNESS | 33.49 | 47.19 | 59.15 | 63.50 | 63.50 | **64.74** |
| 5 | LEGALITY | 44.58 | 46.93 | 58.02 | 60.64 | 60.63 | **64.54** |
| 6 | CRIME | 7.89 | 7.62 | 73.33 | 75.00 | 75.00 | **76.92** |
| 7 | SECURITY | 42.50 | 40.24 | 51.83 | 62.09 | 61.68 | **64.09** |
| 8 | HEALTH | 48.36 | 48.79 | 79.43 | 86.49 | 86.49 | **86.67** |
| 9 | QUALITY OF LIFE | 17.82 | 21.99 | 48.89 | 52.63 | 52.63 | **54.35** |
| 10 | CULTURAL | 15.38 | 15.67 | 51.22 | 52.63 | 52.63 | **55.56** |
| 11 | PUBLIC SENTIMENT | 15.22 | 15.72 | 50.79 | 53.97 | 41.03 | **54.69** |
| 12 | POLITICAL | 49.06 | 48.20 | 50.29 | 46.99 | 46.99 | **47.23** |
| 13 | POLICY | 39.88 | 39.39 | 37.02 | 42.77 | 42.77 | **43.79** |
| 14 | EXTERNAL REGULATION | 12.66 | 14.22 | 44.44 | 66.67 | 66.67 | **71.43** |
| 15 | FACTUAL | 24.64 | 19.21 | 70.95 | 70.37 | 70.41 | **78.95** |
| 16 | PROMOTION | 40.11 | 46.41 | 48.16 | 50.96 | 50.96 | **52.89** |
| 17 | PERSONAL | 45.36 | 46.15 | 59.66 | 62.99 | 62.13 | **71.20** |
| | WEIGHTED AVERAGE | 37.14 | 38.79 | 53.13 | 56.49 | 55.54 | **58.66** |

Table 6: $F_1$ Scores of Unsupervised PSL Models. The highest prediction per frame is marked in bold.

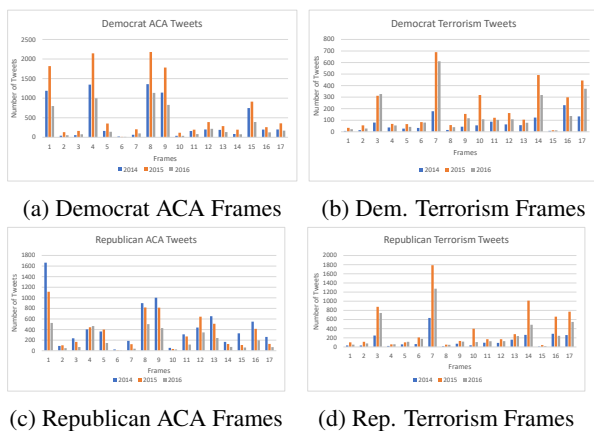

(a) Democrat ACA Frames

(b) Dem. Terrorism Frames

(c) Republican ACA Frames

(d) Rep. Terrorism Frames

Figure 3: Predicted Frames for Tweets from 2014 to 2016 by Party for ACA and Terrorism.

however, use Frame 8 (and Frame 13) to indicate their party's agenda to replace the ACA with access to different options for health care. Additionally, Democrats use the Fairness & Equality Frame (Frame 4) to convey that the ACA gives minority groups a better chance at accessing health care. They also use Frame 15 to express statistics about enrollment of Americans under the ACA. Finally, Republicans use Frames 12 and 13 to bring attention to their own party's actions to "repeal and replace" the ACA with different policies.

Figures 3(b) and 3(d) show the party-based framing patterns over time for terrorism related tweets. For this issue both parties use similar frames: 3, 7, 10, 14, 16, and 17, but to express different views. For example, Democrats use Frame

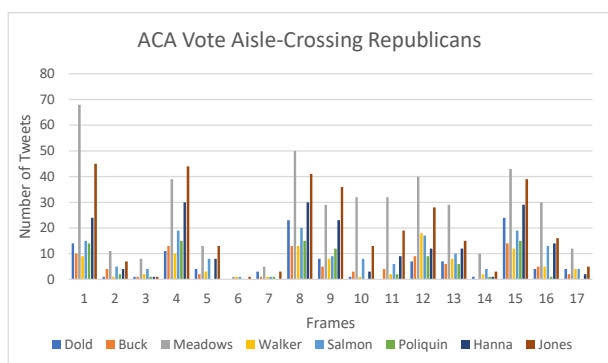

(a) Aisle-Crossing Republicans on ACA Votes.

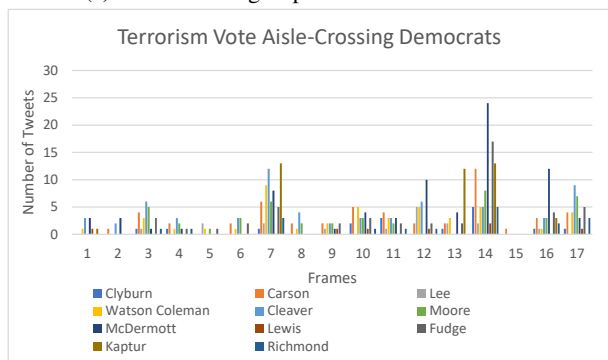

(b) Aisle-Crossing Democrats on Terrorism Votes.

Figure 4: Predicted Frames for Tweets of Aisle-Crossing Politicians in 2015.

3 to indicate moral responsibility to fight ISIS. Republicans use Frame 3 to frame terrorists or their attacks as a result of "radical Islam". An interesting pattern to note is seen in Frames 10 and 14 for both parties. In 2015 there is a large increase in the usage of this frame. This seems to indicate that parties possibly adopt new frames *simultaneously or in response to the opposing party*, perhaps in an effort to be in control of the way the message is delivered through that frame.

**Individual Frames:** In addition to entire party analysis, we were interested in seeing if frames could shed light on the behavior of *aisle-crossing* politicians. These are politicians who do not vote the same as the majority vote of their party (i.e., they vote the same as the opposing party). Identifying such politicians can be useful in governments which are heavily split by party, i.e., governments such as the recent U.S. Congress (2015 to 2017), where politicians tend to vote the same as the rest of their party members. For this analysis, we collected five 2015 votes from the House of Representatives on both issues and compiled a list of the politicians who voted opposite to their party.

The most important descriptor we noticed was that all aisle-crossing politicians *tweet less frequently on the issue* than their fellow party members. This is true for both parties. This behavior could indicate lack of desire to draw attention to one's stance on the particular issue.

Figure 4(a) shows the framing patterns of aisle-crossing Republicans on ACA votes from 2015. Recall from Figure 3 that Democrats mostly use Frames 1, 4, 8, 9, and 15, while Republicans mainly use Frames 1, 8, and 9. In this example, these Republicans are considered aisle-crossing votes because they have voted the same as Democrats on this issue. The most interesting pattern to note here is that these Republicans use the same framing patterns as the Republicans (Frames 1, 8, and 9), but they also use the frames that are *unique to Democrats*: Frames 4 and 15. These latter two frames appear significantly less in the Republican tweets of our entire dataset as well. These results suggest that to predict aisle-crossing Republicans it would be useful to check for usage of typically Democrat-associated frames, especially if those frames are infrequently used by Republicans.

Figure 4(b) shows the predicted frames for aisle-crossing Democrats on terrorism-related votes. We see here that there are very few tweets from these Democrats on this issue and that overall they use the same framing patterns as seen previously: Frames 3, 7, 10, 14, 16, and 17. However, given the small scale of these tweets, we can also consider Frames 12 and 13 to show peaks for this example. This suggests that for aisle-crossing Democrats the use of additional frames not seen in their party might indicate potentially different voting behaviors.

## 7 Conclusion

In this paper we present the task of collective classification of Twitter data for framing prediction. We show that by incorporating Twitter behaviors such as similar activity times and similar networks, we can increase $F_1$ score prediction. We provide an analysis of our approach in both supervised and unsupervised settings, as well as a real world analysis of framing patterns over time. Our global PSL models can be applied to other domains, such as politics in other countries, simply by changing the initial unigram keywords to reflect the politics of those countries.

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
