# Peer review of "Leveraging Behavioral and Social Information for Weakly Supervised Collective Classification of Political Discourse on Twitter"

_ACL 2017 — decision unknown_

[Official Review · Reviewer 1 · rating 4 · confidence 4]
soundness 3 · originality 4 · clarity 5 · impact 4 · substance 4 · appropriateness 5 · meaningful comparison 4 · presentation format Oral Presentation

- Strengths: 1) an interesting task, 2) the paper is very clearly written, easy
to follow, 3) the created data set may be
useful for other researchers, 4) a detailed analysis of the performance of the
model.

- Weaknesses: 1) no method adapted from related work for a result comparison 2)
some explanations about the uniqueness of the task and discussion on
limitations of previous research for solving this problem can be added to
emphasize the research contributions further. 

- General Discussion: The paper presents supervised and weakly supervised
models for frame classification in tweets. Predicate rules are generated
exploiting language-based and Twitter behavior-based signals, which are then
supplied to the probabilistic soft logic framework to build classification
models. 17 political frames are classified in tweets in a multi-label
classification task. The experimental results demonstrate the benefit of the
predicates created using the behavior-based signals. Please find my more
specific comments below:

The paper should have a discussion on how frame classification differs from
stance classification. Are they both under the same umbrella but with different
levels of granularity?

The paper will benefit from adding a brief discussion on how exactly the
transition from long congressional speech to short tweets adds to the
challenges of the task. For example, does past research rely on any specific
cross-sentential features that do not apply to tweets? Consider adapting the
method of a frame classification work on
congressional speech (or a stance classification work on any text) to the
extent possible due to its limitations on Twitter data, to compare with the
results of this work.

It seems “weakly supervised” and “unsupervised” – these two terms
have been interchangeably used in the paper (if this is not the case, please
clarify in author response). I believe "weakly supervised" is
the
more technically correct terminology under the setup of this work that should
be used consistently throughout. The initial unlabeled data may not have been
labeled by human annotators, but the classification does use weak or noisy
labels of some sort, and the keywords do come from experts. The presented
method does not use completely unsupervised data as traditional unsupervised
methods such as clustering, topic models or word embeddings would.  

The calculated Kappa may not be a straightforward reflection of the difficulty
of
frame classification for tweets (lines: 252-253), viewing it as a proof is a
rather strong claim. The Kappa here merely represents the
annotation difficulty/disagreement. Many factors can contribute to a low value 
such as poorly written annotation
guidelines, selection of a biased annotator, lack of annotator training etc.
(on
top of any difficulty of frame classification for tweets by human annotators,
which the authors actually intend to relate to).
73.4% Cohen’s Kappa is strong enough for this task, in my opinion, to rely on
the annotated labels. 

Eq (1) (lines: 375-377) will ignore any contextual information (such as
negation
or conditional/hypothetical statements impacting the contributing word) when
calculating similarity of a frame and a tweet. Will this have any effect on the
frame prediction model? Did the authors consider using models that can
determine similarity with larger text units such as perhaps using skip thought
vectors or vector compositionality methods?  

An ideal set up would exclude the annotated data from calculating statistics
used to select the top N bi/tri-grams (line: 397 mentions entire tweets data
set has been used), otherwise statistics from any test fold (or labeled data in
the weakly supervised setup) still leaks into
the selection process. I do not think this would have made any difference in
the current selection of the bi/tri-grams or results as the size of the
unlabeled data is much larger, but would have constituted a cleaner
experimental setup.  

Please add precision and recall results in Table 4. 

Minor:
please double check any rules for footnote placements concerning placement
before or after the punctuation.

[Official Review · Reviewer 2 · rating 3 · confidence 3]
soundness 3 · originality 4 · clarity 4 · impact 4 · substance 4 · appropriateness 5 · meaningful comparison 4 · presentation format Oral Presentation

- Strengths: The authors address a very challenging, nuanced problem in
political discourse reporting a relatively high degree of success.

The task of political framing detection may be of interest to the ACL
community.

The paper is very well written.

- Weaknesses: Quantitative results are given only for the author's PSL model
and not compared against any traditional baseline classification algorithms,
making it unclear to what degree their model is necessary. Poor comparison with
alternative approaches makes it difficult to know what to take away from the
paper.

The qualitative investigation is interesting, but the chosen visualizations are
difficult to make sense of and add little to the discussion. Perhaps it would
make sense to collapse across individual politicians to create a clearer
visual.

- General Discussion: The submission is well written and covers a topic which
may be of interest to the ACL community. At the same time, it lacks proper
quantitative baselines for comparison. 

Minor comments:

- line 82: A year should be provided for the Boydstun et al. citation

- It’s unclear to me why similar behavior (time of tweeting) should
necessarily be indicative of similar framing and no citation was given to
support this assumption in the model.

- The related work goes over quite a number of areas, but glosses over the work
most clearly related (e.g. PSL models and political discourse work) while
spending too much time mentioning work that is only tangential (e.g.
unsupervised models using Twitter data).

- Section 4.2 it is unclear whether Word2Vec was trained on their dataset or if
they used pre-trained embeddings.

- The authors give no intuition behind why unigrams are used to predict frames,
while bigrams/trigrams are used to predict party.

- The authors note that temporal similarity worked best with one hour chunks,
but make no mention of how important this assumption is to their results. If
the authors are unable to provide full results for this work, it would still be
worthwhile to give the reader a sense of what performance would look like if
the time window were widened.

- Table 4: Caption should make it clear these are F1 scores as well as
clarifying how the F1 score is weighted (e.g. micro/macro). This should also be
made clear in the “evaluation metrics” section on page 6.